# Socio-community care of people with disabilities: Experiences of caregivers living in south-central zone of Chile

Juan Andrés Pino-Morán [1,2,3], Rodrigo González [4,5], Pía Rodríguez-Garrido [2,6,7,8]*,
María Soledad Burrone [1]*

1 Instituto de Ciencias de la Salud, Universidad de O'Higgins, Rancagua, Región del Libertador General Bernardo O'Higgins, Chile, 2 Millennium Nucleus Studies on Disability and Citizenship (DISCA), Santiago, Región Metropolitana, Chile, 3 Grupo de Estudios Críticos de la Discapacidad (CLACSO), Buenos Aires, Argentina, 4 Escuela de Salud, Universidad de O'Higgins, Rancagua, Región del Libertador General Bernardo O'Higgins, Chile, 5 Instituto de Gobierno y Políticas Públicas, Universidad Autónoma de Barcelona, Barcelona, España, 6 Millennium Institute for Care Research (MICARE), Santiago, Región Metropolitana, Chile, 7 Laboratório de Estudos Sociais sobre o Nascimento (Nascer.pt), Instituto Universitário de Lisboa, Lisboa, Portugal, 8 Facultad de Salud y Ciencias Sociales, Universidad de Las Américas, Santiago, Región Metropolitana, Chile

* mariasoledad.burrone@uoh.cl (MSB); pvrodriguez@udla.cl (PRG)

## Abstract

### Introduction

Care involves unequal relationships between gender, corporal conditions and responsibilities. This is especially sensitive for the participation and socio-community inclusion of people with disabilities and their caregivers.

### Aim

To analyze the perceptions of caregivers regarding the construction of socio-community care for people with disabilities in the south-central Zone of Chile.

### Method

Qualitative study from a situated feminist perspective. Ten semi-structured interviews were conducted with women caregivers of people with disabilities during the years 2021–2022. The interviews were based on qualitative thematic analysis.

### Results

The analysis revealed two central categories and six emerging subcategories: a) The world of the system; a.1) Health and rehabilitation, a.2) Education, a.3) Work, and b) The world of life; b.1) Family, b.2) Neighbors and friends, b.3) Community organizations. Socio-community care is a system of action and links that are woven between "The world of the system" and "The world of life" to sustain existence. They are a provision of support and assistance from public institutions and communities.

**Data availability statement:** The revised and final dataset is now available at the following public link: https://doi.org/10.5281/zenodo.17350761.

**Funding:** This research was financed by Agencia Nacional de Investigación y Desarrollo (Fondecyt Postdoctoral N°3220665) and Agencia Nacional de Investigación y Desarrollo (Fondecyt de Iniciación N°11250327). The funders had no role in study design, data collection and analysis, decision to publish, or preparation of the manuscript.The authors received no specific funding for this work.

**Competing interests:** The authors have declared that no competing interests exist.

## Conclusions

Socio-community care is built with a lot of effort and exhaustion on the part of the families of people with disabilities, especially by women. Of particular importance is the responsibility and self-organization of communities to meet the needs and care of persons with disabilities. Hence the urgency to develop comprehensive care systems where the State takes greater and better responsibility.

---

## Introduction

Care is a complex, plural, and polysemic concept, which in recent decades has gained greater attention and position in the global public agenda thanks to the demands of feminist movements [1].

In this sense, care ranges from macro-structural matters, such as environmental care and nature, down to micro-structural affairs, such as caring for people with disabilities. This important fan of comprehension has led to a proliferation of academic studies on the matter [1–3]. The concept is now studied across health, social and political sciences as a vital effort in charge of sustaining daily life and intergenerational reproduction [4,5].

In its variety of perspectives, care is conceived of as an ethical and social problem [6], since it involves unequal relations between genders, body conditions, and responsibilities, apart from not being recognized as a democratically relevant activity [7]; however, this unpaid work [8] shapes our social, affective, and economic relations [4].

Care is not habitually considered from a collective perspective, which is especially sensitive for the participation and socio-community inclusion of people with disabilities and their caregivers [7].

For the purposes of this study, we will define socio-community care as the systems of inter-body links constructed under the principle of reciprocity, respect, and co-responsibility in social and institutional relations established by people within their communities and culture, allowing dialogues, communication, and access for people to services, care and professional help, and not only to traditional family care [9].

Regarding the last point, the social movement of people with disabilities has historically opposed considering care from an exclusively individual, medical, and assistance-based perspective. Personal autonomy and community life have been at the center of their demands. Thus, activism and academia have not been exempt from ethical, political, and epistemic tensions surrounding care and personal assistance [10], which have opened up an important field of research for both feminist studies [1,2,6,11] and critical disability studies [12,13].

Approaching care from a socio-community perspective involves inverting the dominant individualistic, gendered, and family comprehensions towards perspectives which are more situated, corporeal, communitarian, territorial, and integral for national care systems. For this exercise, we will rely on the proposal from Víctor Martinez [14] who has worked on incorporating communities into public programs and policies. Socio-community care organization thus implies analyzing co-responsibilities

in a more collective and inclusive way. This leads us to emphasize the importance of the socio-community element, since it interconnects what we will call "The World of the system" which is the institutional dimension corresponding to the responsibilities of the State and its expression among citizens, with "The World of life" which is the community dimension and responds to communities' own self-organization [14].

Important national care system transformations have occurred recently, particularly in Latin America, giving rise to different public policy action models [3]. This is fundamental to incorporate social and health policies where institutions support popular sectors, which have historically been obliged to self-organize to meet these needs with community initiatives. However, these organizations and initiatives tend to be sustained by a small number of people, usually women from impoverished territories [15]. Care dynamics have thus been affected by the absence of the State and its neoliberal policies, which invisibilize the collective ways of upholding care among and towards people with disabilities [13,16].

Under this logic, we understand that the capitalist, ableist, and heteropatriarchal model hinders collective responsibility. Perez [17] offers this synthesis:

> The socioeconomic system is made for subjects who have neither their own need for care, nor responsibilities for the care of others. Heteropatriarchal capitalism imposes the life objective of transcendence, of self-sufficiency in and through the market. The privileged subject is one which can make it on their own, and can achieve everything with enough effort. If they cannot achieve this, then they fall into the other side of the dichotomy, they are dependent, and their well-being is in the function of the Other. [17], p.208

The neoliberal model in Chile affects the unequal gendered distribution of care. According to Grandon [11] over half of the population living in a dependent situation only receives care from their household members – mainly from a woman – and without external support. Care responsibilities thus fall almost exclusively upon the women of the family [18]. While many people carry out these labors for filial and affective reasons, there is still little community or State co-responsibility.

This situation has been denounced since the 1990s by feminist economics, which warns about the care economy and its relationship with the visibility of domestic and caregiving labor as unpaid work. It also highlights how this generates a significant impact on family, social, and state economies, as it is not recognized as legitimate work, and women end up bearing the highest cost. In this regard, England & Folbre [19] argued that care work creates public goods that contribute to social and community development. Following this argument, caring for people with disabilities contributes to their inclusion in social and economic dynamics, benefiting the entire population.

Similarly, Julie Nelson [20] pointed out the idea of paying for care, at a time when it was believed that this could affect emotional bonds by introducing a supposed commodification of care. From this perspective, the relevance of rethinking the organization of care emerges, including institutional and community support, to move toward more egalitarian and inclusive societies.

In that sense, the relevance of the present study coincides with the policies driven by "The world of the system", through its international institutions, such as the indications from the fifth Sustainable Development objective in the United Nations Development Program (UNDP) which lays out a strategic gender equality program for 2022–2025 raising concern about the gendered division of labor, recognizing the challenges of unpaid care work as a core aspect to incorporate in the public agenda [21]. Similarly, the International Labor Organization (ILO) called for more care investment, providing recommendations for constructing public policies in this regard, since it estimated that 28 million job positions could be created in Latin America in the care sector, mostly formal work, with a transformative investment in care and gender equality policies [8]. This is particularly significant for families in general, as well as for women performing care work who have lost their formal jobs.

The Human Rights Council of the UN High Commission, through its resolution 49/12 of this year, has also discussed the need for support systems to guarantee community inclusion for people with disabilities, among other reasons, as a means to build a better future following the COVID-19 pandemic [21].

In Chile, implementation is currently underway for a National Care System (NCS), with pilot experiences since 2017. This creates an important social policy precedent, since it problematizes and comprehends care in its social, political, and health dimensions, incorporating support and assistance from social and health professionals for people with disabilities as well as their caregivers within their own homes and communities.

Based on all the above, there are experiences of collective and community care that help propose new models for its approach. These experiences generate our interest in analyzing the perceptions of caregivers regarding the construction of sociocommunity care for people with disabilities in the central-southern region of Chile.

## Method

### Design

The framework of the present study is a qualitative research design, which is understood as a situated activity placing the observer within the world of life. It consists of a series of interpretive material practices which make the world visible and transform it [22]. It thus allows for a situated approach which is respectful of the reality to be studied, providing in-depth data, rich interpretation, environmental contextualization, details, and experiences.

In turn, the work was carried out under a situated feminist paradigm in order to reveal the power relations disposed via Eurocentric Western modernity, global capitalism, and colonialism [23–25]. The intention of doing this is to situate the structural position occupied by care for people with disabilities within the O'Higgins region and in Chilean culture.

Orienting the methodology of this study under the indicated focuses allowed us to generate a greater and better approach to the participants, establishing horizontal and fluid dialogues to incorporate their epistemes, contexts, and experiences, which should be the prevailing situation given the adversities involved in carrying out a study with these characteristics on account of the participants' lack of time arising from their care routines.

### Study context

The study was conducted in Chile; a country located in South America and subdivided into 16 regions from north to south. At the time of the interview, all participants lived in the O'Higgins region, which is located in the central-southern part of the country.

To contextualize, in accordance with the Survey on Disability and Dependence 2022 (ENDIDE), the total number of people with disabilities over age 18 in Chile is 2,703,893. Of this number, 115,507 live in the O'Higgins region, which leaves it at 7th place nationwide [26].

The study took place within seven municipalities in the region: San Vicente de Tagua Tagua, Chimbarongo, Lo Miranda, Las Cabras, Rancagua, Olivar, and Peumo, due to our call to action and diffusion within social media as well as because of snowball sampling.

### Participant selection criteria

The objective of the selection of participants was to gather the opinions of people and representatives of organizations that provide informal care for people with moderate-severe disabilities, as well as those who are primary caregivers, who provide care for a period of time longer than 6 months, who do not receive remuneration for caregiving, who are over 18 years of age and who belong to the O'Higgins region.

Potential participants were identified through links with community organizations. Subsequently, a purposive sampling was carried out to identify and invite key informants. From this point, snowball sampling was used, who provided background information on other caregivers [27].

A total of ten participants were interviewed during the years 2021–2022 (see Table 1 and S1 Appendix).

**Table 1. Participants' profile.**

| Interviewees (pseudonyms) | Caregiver role | Condition of the person receiving care |
|---|---|---|
| Jessica | Mother | Moderate dependence |
| Gloria | Daughter | Moderate dependence |
| Maria | Mother | Moderate dependence |
| Emiliana | Daughter | Moderate dependence |
| Elena | Daughter | Moderate dependence |
| Daniela | Mother | Severe dependence |
| Jany | Mother | Moderate dependence |
| Estefania | Daughter | Moderate dependence |
| Pia | Mother | Severe dependence |
| Eugenia | Caregivers' organization representative | Not applicable |

## Information construction techniques

We opted to use semi-structured in-depth interviews, given that, to paraphrase Taylor and Bogdan [28], these encounters are aimed at knowing and understanding the perspective which people have about their own trajectories, life experiences, or life situations in the face of a particular phenomenon or topic.

In this case, the response to the study objective included the participation of nine people carrying out care work and one representative from a community care organization.

All the interviews were done during one session with an average length of one hour. The encounters took place in the caregivers' homes or in nearby places. After notifying the participants, audio recordings were made of the interviews for transcription.

## Ethical considerations

The study was conducted in accordance with the ethical standards set forth in the Helsinki Declaration [29]. Furthermore, the study was approved by the Scientific Ethics Committee of the *Universidad de O'Higgins* in Chile (IRB 017–2022).

The recruitment process for participants was conducted over the course of one year, from July 1, 2022, to July 1, 2023.

Prior to the commencement of the informed consent process, the study objectives were elucidated in exhaustive detail to potential participants. This was followed by a comprehensive delineation of the rationale for participation, the confidentiality and pseudonymity of the data, and the participant's right to withdraw from the study at any point. Once the potential participant had accepted the informed consent, the next step was to sign the document. The consent was obtained in writing, through the participants' signatures.

## Methodological rigor

This study followed the orientations in the protocol 'Enhancing transparency in reporting the synthesis of qualitative research' (ENTREQ) [30], to protect the methodological rigor of the qualitative study.

## Data analysis

Qualitative data analysis was used for the transcribed material; specifically, we considered the six phases proposed by Braun & Clarke [31] which are: a) familiarization with the background and information obtained; b) initial code generation; c) seeking central themes; d) central theme review and selection; e) naming and categorizing the selected themes; and f) final category report. This process was done with Atlas ti version 24 software in Spanish.

The analysis categories were designed based on the theoretical-methodological proposal of the socio-community approach, starting from the design of questions in the interview guide, and then deductively arriving at the final categories. This approach represents an articulated system of connections that provides support networks for people with disabilities and their families.

The analysis identified two central categories: a) "World of the System", which included subcategories a.1) Health and rehabilitation, a.2) Education, and a.3) Work. The second central category identified was: b) "World of Life", identifying three subcategories b.1) Family, b.2) Neighborhood and friends, b.3) Communitary organizations.

## Results

All the participants are female caregivers who have a filial relationship with the individuals with disabilities they care for, and they also live in the same household.

Socio-community care consists of action systems and ties which are woven between "The World of the system" and "The world of life" to uphold existence. This means that socio-community care is a disposition of support and assistance from public institutions and communities for development of people in their territories. This framework particularly emphasizes communities' responsibility and self-organization to handle the needs and care of people with disabilities.

The above categories and subcategories are closely related to the notion of "operant network" and "non-operant network", insofar as the former represents the facilitating aspects and the latter the hindering aspects of caregiving.

The next part presents the most relevant findings, indicating the barriers and facilitators faced by caregivers in community care, creating a differentiated way of living through care work.

a) World of the System

This "world" is fundamental for weaving care within its institutional dimension. These examples indicate the impact on caregivers' trajectories when institutional networks are operational, contributing to wellbeing and strengthening interpersonal bonds, and by contrast, when they are inoperative and generate barriers to caregivers' participation and wellbeing (Table 2).

**Table 2. Networks in the World of the System.**

| Networks | Operating Network | Inoperative Network |
|---|---|---|
| **a.1) Health and rehabilitation** | *And they go there every Tuesday [public rehabilitation center]. And it's like my day off. It's like my free morning. (a.1.1 Jessica)*<br><br>*We talked [with health professionals], because they saw the situation there was. I started trusting them and letting things out to them, because I felt bad too (a.1.2 Gloria)* | *What happens is that they don't come, they don't come any more and if there's an hour or something, there still isn't really anything, because I have to go, get up early, be in around seven at the poly (health center) and line up. (a.1.3 Maria)*<br><br>*She isn't in the prostrate program, because I got her up. Just because I get her up from the chair, she isn't in the dependency [program]. (a.1.4 Pia)* |
| **a.2) Education** | *There was 100% disposition, they have a nursing technician, who's a wonderful gal. There's physical therapy, neurology, occupational therapy, everything at the school. (a.2.1 Maria)*<br><br>*She [the job workshop professional] makes me calm when I leave them. I know that if there's an emergency, she can get them out of it, and knows how to regulate it. (a.2.2 Daniela)* | *There was so much negativity from the school that took XXX in, that at the end of the year we wound up filing a complaint with the Superintendency. The next year they finally had all the resources for him. (a.2.3 Jessica)*<br>*Education isn't a support in the transition to adult life, it's not help, and so it's a burden for you. (a.2.4 Daniela)* |
| **a.3) Work** | *Yes, I created an Instagram store, so that way I can work and pretty much handle things (a.3.1 Daniela).*<br><br>*So he has a business making food at home and distributing it. The hours are short and we try to fit everything into the kids' schedules, but like I say, I stay with XXX and take care of her all day. (a.3.2 Jany).* | *I was a packing supervisor and moved around. But now I had to leave that. My sister worked with me, so we had to quit work to care for Dad. (a.3.3 Elena).*<br><br>*So it's sort of more like a normal job. Like the jobs I've had, where you leave around 1700 or 1800 hours. Then there was time available, for what you want to do. But it's not like that here, because you always have to keep watching (a.3.4 Emiliana).* |

These barriers may arise due to the lack of institutional presence in the area where the caregivers and individuals with disabilities live, which often occurs in rural areas, or also due to the lack of programs that meet their support and assistance needs.

a.1) Health and rehabilitation

One of the important spaces for caregivers is the support network they receive and/or expect to receive from the health and rehabilitation sectors, as the needs of the individuals they care for require professional guidance. These sectors provide the main directions for caregiving. However, when individuals become more dependent, the healthcare system begins to neglect this user profile, which ends up overloading the caregivers and highlighting a disconnection between institutions and the reality of the caregivers, often resulting in loneliness and institutional neglect (a.1.3).

This is an indication of the barriers faced by caregivers and people with disabilities to access the right to health care, mainly from the health network management and the available disability care supply especially in the primary health sector. Apropos of this situation, Chile has a law on preferential caregiver access (Law #21,380), which as we can note has not yet become very socialized within healthcare systems (a.1.3). We can also observe how the health system has an ableist support provision perspective, reflected in the profiles of caregivers receiving economic support, which is based on people being fully prostrate, which is ultimately iatrogenic as an incentive, as it promotes the loss of functionality in order to receive economic benefits, especially among the most impoverished populations (a.1.4).

However, when health and rehabilitation networks are operational and provide the necessary support to caregivers and individuals with disabilities, they generate more satisfactory experiences. In this way, healthcare becomes a protective factor for the caregiving dynamic, not only as a provision of health for the individual with a disability, promoting their recovery and rehabilitation processes, but also reducing the psychological and labor burden on caregivers (a.1.1), offering opportunities to explore and participate in other activities, providing well-being and relief in their daily routines (a.1.2). This is like the previous example, which highlights the strategy of Primary Health Care, implemented by Community Rehabilitation Centers, which are local, public health institutions committed to providing well-being and health participation to areas with larger rural populations in Chile.

a.2) Education

Education is a human right for all people. It enables holistic development as citizens. However, when educational institutions are not prepared to guarantee access for individuals with disabilities, it becomes a real struggle to find institutions with the necessary conditions for the educational and formative process that individuals with disabilities require. In this process, the burden and concern mainly fall on the family, particularly on the caregivers (a.2.3).

Accessibility barriers to educational participation are framed as an exclusion problem for individuals with disabilities and their caregivers, as there is no suitable space or teacher preparation to support transitions to independent living. This also creates a greater social and psychological burden for caregivers and families, where often the caregivers themselves end up fulfilling educational roles in the absence of spaces tailored to their children's needs (a.2.4).

However, when operational networks exist, satisfactory education inclusion experiences are achievable, once the resources and professionals needed for integral work are present. These situations help decompress the care hours which many mothers of children with disabilities must handle at home (a.2.1).

We can also observe the importance of caregivers having a space where they feel supported, a safe caregiving space for both themselves and their children. This is called an operational action system, where state institutions can make a significant positive impact. This has a positive future repercussion, as, to the extent that individuals with disabilities can access education, they will eventually be able to integrate into the workforce (a.2.2).

a.3) Work

Work plays a fundamental role in recognizing individuals as citizens. However, when it comes to people with disabilities, the absence of productive capabilities is often assumed, promoting a negative and discriminatory social representation. Similarly, caregiving work is also seen as unproductive labor, being culturally and economically invisible to society.

Taking on family care inevitably involves a rupture with formal institutional work networks to which caregivers belonged. Once care work begins, one of the most immediate ruptures is with labor networks, where the person who begins providing care must abandon their production labor and take on a different type which receives no social or monetary recognition *(a.3.3)*.

Care involves a responsibility, concern for another, sustained physical and mental effort over time; the action of care mobilizes and facilitates productive flow. However, taking on a caregiver role and abandoning institutional work networks has a strong impact on the economic autonomy of caregivers and their families, who mostly fall into poverty *(a.3.4)*.

There are experiences of efforts to reconstruct work itineraries which involves a major family and personal effort to generate new micro-economies to support and provide family sustenance. However, it cannot be ignored that these efforts require more state support, granting more and better protection, in order to not ultimately create more concern and overburdening for caretakers and their families. Small businesses must have guarantees and supports to allow for sustainable care work *(a.3.1) (a.3.2)*.

b) World of Life

"The world of life" provides a sense of belonging to and identifying with a community, and shows the possibility of weaving territorial autonomy to create co-responsibility for the situation of all its members, regardless of their corporeal condition and/or ability (Table 3).

**Table 3. Networks of the World of Life.**

| Networks | Operating Network | Non-Operating Network |
|---|---|---|
| **b.1) Family** | *Well, family's been super important. My daughter and her kids have always been there to help us out. We have a really good support network (b.1.1 Fabiola).*<br><br>*I have two other sisters, and they know how to feed her too. They've had to stay with her too. So they know. (b.1.2 Maria)* | *I still find it sort of, uh, mean. That is, it's always us women who have to take responsibility (b.3.3 Alejandra).*<br><br>*We broke up when XXX was four, and he's absent. He left the mom and he left everything behind (b.4.4 Maria).* |
| **b.2) Neighbors and friends** | *My husband has a friend. And that friend takes him to the soccer field, because my son loves to go there. (...) And he says to him, hey, ''You know what?'' I'm going to come for XXX to take him to the field. And he comes. (b.2.1 Jessica)*<br><br>*In fact, when my dad got sick, the neighbors asked after him every day. They came to ask if he needed something, or if we had to leave, they'd mind the house. (b.2.2 Elena)* | *Look, with my neighborhood I don't have, let's say, a lot of closeness if I go around there. Some neighbors know what my daughter has, not all of them, because my neighborhood isn't a good one. (b.2.3 Jany)*<br><br>*Yes, because with people, you can't [say] just anything, because after you say something to that person, they add on something else, a different thing, something or other that it isn't. (b.2.4 Gloria)* |
| **b.3) Community organizations** | *Fellowship, putting yourself in someone else's shoes. Everyone always lends a hand. (b.3.1 Jany)*<br><br>*Coming into an association where your peers have the same troubles, you speak the same language with them. It was really powerful; we started building each other up. (b.3.2 Daniela)* | *But for the same reason, for XXX, they always had meetings when I was with him. So, I know him, he'll get bored anyway and be with him, so wait, I'd rather not come in because I won't be paying 100% attention to what they're saying. (b.3.3 Jessica)*<br><br>*But it was hard, because here people are different from Santiago anyway. Because people think that kids like XXX are only visible in the ''Teletón'' for the ''27 hours of love.'' And they almost think that what they have is contagious. (b.3.4 Jessica)* |

b.1)  Family:

One crucial link in the community care chain is the structure and functioning of families. However, this care chain is constructed based on feminization, which perpetuates a patriarchal and ableist care model *(b.3.3)*. This is reflected in the scarcity of men involved in caregiving for individuals with disabilities, but also in the assumption that women are the only ones with the innate ability to care. This representation of care has such deep cultural roots that when these inequalities are problematized, it is ultimately justified with the traditional gendered division of labor and the male provider role *(b.4.4)*.

We can thus comprehend the importance of moving forward with care chains which are not always entirely and inescapably within the private and family sphere. One step in this transformation is to recognize responsibility in care tasks, and not to naturalize them as feminine labor *(b.1.2)*. In this way, children, parents and partners can take on care work as part of a culture of non-feminized care *(b.1.1)*.

b.2)  Neighbors and friends:

The formation and social organization of the neighborhood and friendships are vital, both for social recognition and for support in the distribution of caregiving. Important relations arise from the daily routines of people, within their natural systems, and particularly in their neighborhoods and vicinities. However, within current sociodemographic transitions of the rural-to-urban shift and gentrification of rural sectors, it becomes difficult to form such relations, especially in spaces which are perceived as unsafe *(b.2.3) (b.2.4)*, ultimately limiting the construction of strengthened communities.

There is also a large difference when there is more cohesion within neighborhoods where community processes are identified and recognized *(b.2.2)*. One example of greater co-responsibility can be noted in trust-building experiences to allow accompaniment by people from beyond families, such as friends of the caregiving family *(b.2.1)*. This type of support via friendship helps distribute care work, and offers a more accessible option to activate committed, collective care within territorial communities themselves. This relieves the caregivers' time, but it also reflects the knowledge and emotional involvement of the communities.

b.3)  Communitary organizations:

We can highlight the relevance of community care organizations as an example for constructing territorial mutual care policies. However, the difficulty in accessing these organizations translates into time barriers and resource centralization. This is especially evident in rural areas, where the availability of services and spaces for socialization is lower than in urban contexts *(b.3.3)*. This situation leads to frustration and indifference, increasing sensations of loneliness. These phenomena are common in rural contexts where care work is still established within private and family spheres, and with negative cultural representations associated with disability *(b.3.4)*.

There is also evidence for the importance of organizational support, which is social, emotional, and economic. Organizations can articulate and direct collective efforts to fill health needs *(b.3.1)*.

The role which organizations fulfill implies a nexus between the "world of life" and the "world of the system", as they have more representation in creating public care policies. In this way, they allow for greater cohesion and operativity in the fabric of social and community networks *(b.3.2)*.

## Discussion

The findings of this study invite a rethinking of sociocommunal care as a form of situated interdependence, in which the tension between institutional practices and community-based practices shapes everyday forms of care citizenship. This interpretation aligns with the ethics of care and with feminist and decolonial approaches that shift individualistic perspectives toward relational and contextual understandings of well-being [6,13,23,25]· as well as with community-based frameworks that articulate public and collective responsibilities [14].

In the fields of health and rehabilitation, international evidence consistently describes the challenges and facilitators involved—such as discontinuities in care, geographical distance, eligibility criteria, and communication failures—alongside protective practices that emerge when teams and devices are person-centered [21,32,33] This body of research resonates with human rights frameworks that position community inclusion and support systems as central to post-pandemic reconstruction [34]. Taken together, these contributions suggest that the key is not merely to expand service provision, but to ensure territorial and communicative agreements that sustain dignified and predictable care trajectories.

In the field of education, the literature emphasizes that effective inclusion depends on intersectoral coordination and community-based supports for transitions to adult life; otherwise, the school system may reproduce burdens on families [35,36]. This perspective shifts the focus from individual "adaptation" toward the construction of support ecologies that connect schools, health services, and local networks, preventing educational participation from depending on private resources or the invisible availability of family time.

In the field of labor and the care economy, numerous studies document the adverse effects of informal caregiving on labor market participation, health, and social involvement [37,38], as well as the transfer of unpaid economic value to the system [39]. From a feminist political economy perspective, the public good nature of care [19] and the legitimacy of its remuneration [20] support policies that recognize care as productive work, value its macro-social contribution, and finance productive initiatives anchored in local territories.

Family organizations remain the primary support structure for care in Latin America, following a pattern of feminization that naturalizes women's responsibility and displaces social co-responsibility [40,41,42]. Several authors have problematized the moral and political burden this entails [43], and have highlighted the temporal and material precarization processes faced by caregivers [44,45]. Strengthening a culture of co-responsibility requires acknowledging the plurality of family organizations and avoiding policies that reinforce the privatization of care within households.

Neighborhood and community networks emerge in the literature as spaces capable of sustaining collective forms of care amid successive neoliberal reforms, even in contexts of poverty [46]. Territorial devices—such as neighborhood workers and local organizational experiences—have been analyzed as de facto care policies that challenge dominant meanings and expand everyday citizenship [7,15,47]. However, their sustainability depends on conditions of safety, social infrastructure, and institutional recognition.

Civil society organizations play a bridging role between communities and institutions by translating needs, mobilizing resources, and politically representing groups in situations of vulnerability [48]. The literature on community-based approaches and on the regional development of support and care systems converges in emphasizing that their participation is essential to move from fragmented programmatic logics toward comprehensive care architectures [3,14].

From a conceptual standpoint, debates on autonomy and dependency invite an understanding of autonomy as a relational capacity to build and govern supports, rather than as their absence [49–51]. This perspective reconnects discussions on disability, gender, and care citizenship with approaches that conceive care as a public practice rather than merely a private household duty [7].

From this body of evidence, coherent policy implications can be derived: a) Aligning eligibility criteria and benefits with functional gains and independent living; b) Financing stable community-based supports in health and education; c) Recognizing and remunerating care work and the territorial micro-economies that sustain it; and d) Consolidating intersectoral coordination mechanisms with local governance.

These orientations converge with international frameworks on investment in care and decent work [8], and with regional proposals for building support and care systems oriented toward community living [3,14,34].

## Conclusions

Sociocommunitarian care is built with great effort by the families of individuals with disabilities, particularly by women, which makes the development of integrated care systems even more urgent to extend responsibility to communities and

the state. One proposal is to advance in coordinating existing networks in the "World of the system" with networks in the "World of life." This is particularly relevant to prevent the institutionalization of individuals with disabilities and close the gender-based institutionalization gaps in caregiving, as both situations affect the community life of both caregivers and individuals with disabilities. Therefore, it is essential to understand caregiving systems beyond the domestic and private sphere of the family, in order to extend shared responsibility and recognition to neighborhoods and community organizations for individuals with disabilities and caregivers, as seen in this study. These factors affect the quality of life not only of the caregiver-disabled person dyad but also of the communities in general.

## Supporting information

**S1 Appendix. Interview Guide Caregiver.**
(DOCX)

## Acknowledgments

This work was financed by the Publications Support Fund of the Universidad de O'Higgins.

We would like to express our sincere gratitude to all the individuals who participated in our study.

## Author contributions

**Conceptualization:** Juan Andrés Pino-Morán, Rodrigo González, María Soledad Burrone.

**Formal analysis:** Juan Andrés Pino-Morán, Rodrigo González, María Soledad Burrone.

**Funding acquisition:** Juan Andrés Pino-Morán.

**Investigation:** Juan Andrés Pino-Morán.

**Methodology:** Pía Rodríguez-Garrido.

**Project administration:** Juan Andrés Pino-Morán.

**Resources:** Juan Andrés Pino-Morán.

**Supervision:** Juan Andrés Pino-Morán, María Soledad Burrone.

**Validation:** Juan Andrés Pino-Morán, Rodrigo González, Pía Rodríguez-Garrido, María Soledad Burrone.

**Visualization:** Juan Andrés Pino-Morán, Rodrigo González, Pía Rodríguez-Garrido, María Soledad Burrone.

**Writing – original draft:** Juan Andrés Pino-Morán, Rodrigo González, Pía Rodríguez-Garrido, María Soledad Burrone.

**Writing – review & editing:** Juan Andrés Pino-Morán, Rodrigo González, Pía Rodríguez-Garrido, María Soledad Burrone.

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
