## [Decision Letter · Decision Letter 0]

1 Oct 2024

Dear Dr. Burrone,

We look forward to receiving your revised manuscript.

Kind regards,

Rafael Galvão de Almeida, PhD.

Academic Editor

PLOS ONE

Journal Requirements: When submitting your revision, we need you to address these additional requirements. 1. Please ensure that your manuscript meets PLOS ONE's style requirements, including those for file naming. The PLOS ONE style templates can be found at https://journals.plos.org/plosone/s/file?id=wjVg/PLOSOne_formatting_sample_main_body.pdf and https://journals.plos.org/plosone/s/file?id=ba62/PLOSOne_formatting_sample_title_authors_affiliations.pdf 2. Thank you for stating the following financial disclosure: "This research was financed by ANID-Fondecyt Postdoctoral N°3220665."  Please state what role the funders took in the study.  If the funders had no role, please state: ""The funders had no role in study design, data collection and analysis, decision to publish, or preparation of the manuscript."" If this statement is not correct you must amend it as needed. Please include this amended Role of Funder statement in your cover letter; we will change the online submission form on your behalf. 3. In the online submission form, you indicated that "Data are available upon request as the study has not been completed. These can be requested from the principal investigator, first author of this article." All PLOS journals now require all data underlying the findings described in their manuscript to be freely available to other researchers, either 1. In a public repository, 2. Within the manuscript itself, or 3. Uploaded as supplementary information.This policy applies to all data except where public deposition would breach compliance with the protocol approved by your research ethics board. If your data cannot be made publicly available for ethical or legal reasons (e.g., public availability would compromise patient privacy), please explain your reasons on resubmission and your exemption request will be escalated for approval.  4. Please include your full ethics statement in the ‘Methods’ section of your manuscript file. In your statement, please include the full name of the IRB or ethics committee who approved or waived your study, as well as whether or not you obtained informed written or verbal consent. If consent was waived for your study, please include this information in your statement as well.

Reviewers' comments:

**Comments to the Author**

1. Is the manuscript technically sound, and do the data support the conclusions?

Reviewer #1: Yes

Reviewer #2: Yes

2. Has the statistical analysis been performed appropriately and rigorously?

Reviewer #1: N/A

Reviewer #2: N/A

3. Have the authors made all data underlying the findings in their manuscript fully available?

Reviewer #1: Yes

Reviewer #2: Yes

4. Is the manuscript presented in an intelligible fashion and written in standard English?

Reviewer #1: Yes

Reviewer #2: Yes

Reviewer #1: Comments to the author:

The article proposes to analyze the perceptions of caregivers regarding the construction of socio community care for people with disabilities in the south-central Zone of Chile, using a qualitative approach from a situated feminist perspective. Both the justification for the importance of the study and the objectives are clear. In addition, the article is presented in an intelligible fashion and is written in standard English.

My main criticism is that the study is decontextualized from the original theoretical discussion about unpaid care labor – in the context of the so-called "Care Economics" – which began in the 1990s in the United States. See, for example, some pioneering works in the area such as:

England, Paula and Folbre, Nancy. 1998. “The Cost of Caring.” The Annals of the American Academy of Political and Social Science 561, 39-51

Himmelweit, Susan. 1996. “Conceptualising Caring”. Paper presented at the International Association for Feminist Economics Conference, Washington D.C., July 1996

Nelson, Julie A. 1999. “Of Markets and Martyrs: Is it OK to Pay Well for Care?” Feminist Economics 5: 3, 1-17

In the Discussion section (pp. 14-16), a preliminary and cursory literature review on the topic of care, particularly in the context of caring for disabled individuals, is presented. I propose a reorganization of the article to include a separate section for contextualization and a comprehensive literature review on unpaid care labor immediately following the introduction. This structural adjustment will facilitate a more coherent analysis of the research findings in relation to existing theoretical studies.

Reviewer #2: I suggest detailing the methodology more clearly, what design was used. In addition, I suggest organizing the methodology sections in a more logical way.

The “context of the study” section seems unrelated to the rest. I suggest making the link. The same with table 1, where the column (% of rural population) seems more like an isolated background, unless it is integrated into the story. Basically, answering the question: why are both pieces of information important?

Does the theoretical approach chosen for the analysis have any contribution other than separating institutional and community care? This division is quite classic when carrying out support analysis, so I wonder if the author and theoretical approach chosen could contribute something other than that division.

Regarding the first part of the results, I suggest explaining in the methodology section how the internal categories were established for the two predetermined categories.

Regarding the results, very general statements are made that would benefit from integrating more data or examples. An example is the allusion to the fact that care in the family is feminized and ableist. However, there is no detailed discussion with data on what this means and how it is observed. In most of the categories, this format is replicated, which makes the depth of the data and the richness of the results as a contribution disappear.

I consider that the discussion requires some attention. Although its content is relevant to the topic, it is presented more as background than as a discussion of the findings. Some sections can be written as an introduction, without the results. I suggest going into more depth so that the findings are clearly reflected.

In the conclusions, it is established that the integration of both networks reduces institutionalization. This seems to be new information that has not been presented or analyzed before, and due to the characteristics of a conclusion, I consider that it should not be presented in a new and general way there. It is an interesting dimension that should be addressed in the discussion, if the findings lead in that direction.

In general, rurality does not seem to be relevant to the study, despite the fact that it is presented in the title and as an inclusion criterion. There is nothing on this aspect even in the bibliography, and it seems that it was not treated as a relevant topic in the interview questions. Was it? If this was relevant to the analysis, I suggest incorporating it. If not, I suggest considering it.

**Do you want your identity to be public for this peer review?** For information about this choice, including consent withdrawal, please see our Privacy Policy

Reviewer #1: No

Reviewer #2: No

---

## [Author Response · Author response to Decision Letter 1]

29 Aug 2025

Attached is a table of responses to editors and reviewers in "attach files"

In accordance with the email received on June 4, 2025, we were instructed to resubmit the manuscript without track changes. Therefore, we are attaching the document as requested.

---

## [Decision Letter · Decision Letter 1]

17 Sep 2025

Dear Dr. Burrone,

We look forward to receiving your revised manuscript.

Kind regards,

Rafael Galvão de Almeida, PhD.

Academic Editor

PLOS ONE

Journal Requirements:

Reviewers' comments:

Reviewer's Responses to Questions

**Comments to the Author**

Reviewer #1: All comments have been addressed

Reviewer #2: (No Response)

2. Is the manuscript technically sound, and do the data support the conclusions?

Reviewer #1: Yes

Reviewer #2: Partly

3. Has the statistical analysis been performed appropriately and rigorously?

Reviewer #1: N/A

Reviewer #2: N/A

4. Have the authors made all data underlying the findings in their manuscript fully available?

Reviewer #1: Yes

Reviewer #2: No

5. Is the manuscript presented in an intelligible fashion and written in standard English?

Reviewer #1: Yes

Reviewer #2: Yes

Reviewer #1: In the previous round of review, my main criticism was that the paper's results were decontextualized from the original theoretical discussion about unpaid care labor which began in the 1990s in the United States. I proposed a reorganization of the article to include a separate section for contextualization and a comprehensive literature review on unpaid care labor immediately following the introduction. The idea was that, in so doing, a coherent analysis of the research findings in relation to existing theoretical studies would be possible. The authors adreesed properly both issues.

Re-reading the text after the review, it occurred to me that the details provided by the methodology may not be sufficient to allow the research to be reproduced elsewhere. Therefore, I will also suggest the authors to publish the questionnaire used in the survey as an appendix.

Reviewer #2: Most of the comments have been appropriately addressed. However, the relationship between results and discussion remains weak and needs to be strengthened. The statement regarding rurality remains incomplete. Additionally, the data are not available at the indicated DOI.

**Do you want your identity to be public for this peer review?** For information about this choice, including consent withdrawal, please see our Privacy Policy

Reviewer #1: No

Reviewer #2: No

---

## [Author Response · Author response to Decision Letter 2]

27 Nov 2025

Dear reviewers, we sincerely appreciate the constructive suggestions made on the manuscript. All the suggested revisions have been carefully addressed.

---

## [Decision Letter · Decision Letter 2]

8 Dec 2025

Socio-community care of people with disabilities: Experiences of caregivers living in south-central Zone of Chile

PONE-D-24-35334R2

Dear Dr. Burrone,

We’re pleased to inform you that your manuscript has been judged scientifically suitable for publication and will be formally accepted for publication once it meets all outstanding technical requirements.

Kind regards,

Rafael Galvão de Almeida, PhD.

Academic Editor

PLOS One

Additional Editor Comments (optional):

Reviewers' comments:

Reviewer's Responses to Questions

**Comments to the Author**

Reviewer #1: All comments have been addressed

2. Is the manuscript technically sound, and do the data support the conclusions?

Reviewer #1: Yes

3. Has the statistical analysis been performed appropriately and rigorously?

Reviewer #1: Yes

4. Have the authors made all data underlying the findings in their manuscript fully available?

Reviewer #1: Yes

5. Is the manuscript presented in an intelligible fashion and written in standard English?

Reviewer #1: Yes

Reviewer #1: All the previous comments and suggestions have been fully addressed. The paper is ready for publication.

**Do you want your identity to be public for this peer review?** For information about this choice, including consent withdrawal, please see our Privacy Policy

Reviewer #1: No

---

## [Editor Report · Acceptance letter]

PONE-D-24-35334R2

PLOS One

Dear Dr. Burrone,

I'm pleased to inform you that your manuscript has been deemed suitable for publication in PLOS One. Congratulations! Your manuscript is now being handed over to our production team.

Kind regards,

on behalf of

Dr. Rafael Galvão de Almeida

Academic Editor

PLOS One